# Impact of Foot Surgery and Pharmacological Treatments on Functionality and Pain in Rheumatoid Arthritis: A Five-Year Longitudinal Study

**DOI:** 10.3390/healthcare13091004

**Published:** 2025-04-27

**Authors:** Amparo Campos-Cano, Alejandro Castillo-Dominguez, Ana-Belen Ortega-Avila, Laura Ramos-Petersen, Gabriel Gijon-Nogueron, Maria-Jose Perez-Galan, Andres Reinoso-Cobo

**Affiliations:** 1Department of Nursing and Podiatry, Faculty of Health Sciences, University of Malaga, Arquitecto Francisco Peñalosa 3, Ampliación de Campus de Teatinos, 29071 Malaga, Spain; amparocampos@uma.es (A.C.-C.); alejandrocastillo@uma.es (A.C.-D.); anaortavi@uma.es (A.-B.O.-A.); gagijon@uma.es (G.G.-N.); andreicob@uma.es (A.R.-C.); 2IBIMA Plataforma BIONAND, 29010 Malaga, Spain; 3Department of Rheumatology, Hospital Universitario Virgen de las Nieves, 18014 Granada, Spain; mariaj.perez.galan.sspa@juntadeandalucia.es

**Keywords:** rheumatoid arthritis, foot function, surgery, methotrexate, biologic therapy, disease progression, Hallux abductus valgus

## Abstract

Background: Rheumatoid arthritis (RA) frequently leads to foot deformities, significantly impacting pain, mobility, and quality of life. Surgical and pharmacological treatments are prescribed to manage symptoms, but their long-term effects on foot function remain unclear. This study evaluates the impact of different treatment approaches, including surgery, methotrexate (MTX), and biological therapy (Bio), on foot functionality and pain progression over five years. Methods: A longitudinal cohort study was conducted with 103 RA patients classified into five groups: surgery, MTX < 10 years, MTX ≥ 10 years, Bio < 10 years, and Bio ≥ 10 years. Data from 2018 and 2023 were compared using the Visual Analog Scale (VAS), the Manchester Foot Pain and Disability Index (MFPDI), and the Foot Function Index (FFI). Statistical analyses included ANOVA, Kruskal–Wallis, and ROC curve analysis to assess differences between groups and identify key progression factors. Results: Patients with ≥10 years of disease duration and non-biological treatment (MTX ≥ 10 years) experienced the most severe deterioration in foot function, with a mean FFI increase of +11.89 points (*p* < 0.01). In contrast, MTX < 10 years was the only group to show an improvement in foot function (FFI: −5.29, *p* = 0.02). The surgery group exhibited moderate but highly variable functional changes, while patients on biological therapy showed less progression in pain and disability compared to their non-biologic counterparts. Hallux abductus valgus severity increased across all groups. Conclusions: Patients with long-standing RA on non-biologic therapy exhibited the greatest decline in foot function, whereas early treatment with MTX (<10 years of disease duration) appeared to slow deterioration. Surgery did not consistently provide functional benefits, and biologics helped mitigate progression, though outcomes varied. These findings underscore the importance of early intervention and personalized treatment strategies for foot preservation in RA.

## 1. Introduction

Rheumatoid arthritis (RA) is a disorder characterized by chronic inflammation of the joints, leading to significant structural changes [1]. Foot deformity is one of the main manifestations of RA, affecting more than 90% of patients as the disease progresses [2]. The most common foot deformities in patients with RA primarily affect the forefoot and hindfoot, including conditions such as Hallux Abductus Valgus (HAV), dorsal subluxation of the lesser metatarsophalangeal joints (MPJ), metatarsus primus varus, hallux rigidus, hindfoot valgus, and flatfoot [3,4]. Foot problems in patients with RA affect gait, reduce physical activity and quality of life, and can impair self-image due to decreased mobility and functional capacity. This limits social participation and, consequently, negatively impacts quality of life [5,6].

Currently, RA relies on disease-modifying antirheumatic drugs (DMARDs), which include conventional synthetic DMARDs (csDMARDs), with methotrexate (MTX) as the leading agent, as well as targeted therapies, comprising biologic DMARDs (Bio) and targeted synthetic DMARDs (tsDMARDs). The most widely accepted therapeutic strategy is the treat-to-target (T2T) approach, which aims to achieve and maintain early clinical remission or low disease activity to slow radiographic progression and reduce the risk of systemic complications. Despite significant advancements in RA management, a definitive cure remains elusive, and a subset of patients fails to achieve remission or adequate disease control. These cases pose a considerable therapeutic challenge, highlighting the need for ongoing research into novel strategies to optimize clinical outcomes and improve patients’ quality of life [7].

On the other hand, there are non-pharmacological treatments, such as surgery. Surgery is indicated when conservative treatment is insufficient; it alleviates pain caused by synovitis and joint destruction, correcting deformities, optimizing gait patterns, and adapting the foot to footwear [8,9]. The combination of arthrodesis of the first MPJ and resection arthroplasty of all the lesser metatarsal heads has historically been recognized as the gold standard treatment for rheumatoid forefoot deformities [10,11,12]. However, complications have been reported, such as recurrence of HAV [13,14], hammertoe deformity, and painful callosities [15]. Therefore, joint-preserving techniques are now increasingly being adopted [9,16,17,18]. This approach combines performing osteotomies on the metatarsals with soft tissue reconstruction, aiming to correct deformities while preserving joint function [19,20].

The objective of this study is to compare the effects of foot surgery in RA patients with those undergoing MTX and biologic treatments, categorized by disease duration. Foot functionality, pain, mobility, and deformity are assessed based on measurements taken in 2018 and 2023 to determine their impact on patients’ quality of life and well-being.

## 2. Materials and Methods

### 2.1. Design and Ethical Approval

This was a cohort follow-up study of participants. This study received ethical approval from the Portal de Ética de la Investigación Biomédica de Andalucía (PEIBA), which authorized and expanded it into a longitudinal study (ARC0001). The study was conducted in full compliance with the principles of the Declaration of Helsinki for ethical medical research involving human participants and was approved by the Ethics Committee.

### 2.2. Participants

In 2018, the first part of the study was initiated with a cohort of 623 patients diagnosed with RA, with the aim of analyzing the process of foot deformation. All met the 2010 classification criteria for AR established by the American College of Rheumatology and the European League Against Rheumatism [21]. Later, in 2023, continuing studying the same cohort, 20 patients who had undergone foot surgery for the first time were identified with similar disease progression to non-operated patients. They were compared with 4 other groups defined according to pharmacological treatment and time of evolution. Patients who changed their pharmacological treatment, underwent lower limb surgery, or presented dementia or inability to walk were excluded, therefore not being selected to be part of one of the four control groups. The codes that were assigned to the participants in 2018 were used to randomly select the patients in 2023 after exclusion. This process was generated with SPSS statistics, allowing the control of confounding factors and a more homogeneous distribution.

The study groups are as follows:Group 1 (Surgery): patients who underwent forefoot surgery.Group 2 (MTX < 10 years): patients treated with methotrexate with a disease duration of less than 10 years.Group 3 (MTX ≥ 10 years): patients treated with methotrexate with a disease duration of 10 years or more.Group 4 (Bio < 10 years): patients receiving biological therapy with a disease duration of less than 10 years.Group 5 (Bio ≥ 10 years): patients receiving biological therapy with a disease duration of 10 years or more.

In the study, participants were classified by pharmacological group according to their mechanism of action and use in RA, which is a more representative approximation of clinical practice.

The cutoff point was set between subgroups less than or greater than 10 years for convenience, seeking a direct comparison between the surgery group and the subgroups by years of evolution and differentiating them according to treatment [22].

Exclusion criteria included patients with other systemic diseases, prior surgeries unrelated to RA, or difficulty/inability to complete questionnaires and/or interviews. Additional exclusions were the inability to ambulate or move, total or partial lower limb amputation, and surgeries unrelated to trauma.

### 2.3. Demographic and Clinical Characteristics

The following demographic information was collected: age, gender, years of disease progression, pharmacological treatment, and surgical procedures. Clinical data were collected using the Visual Analog Scale (VAS) for both general and foot-specific pain, ranging from 0 to 10, where 0 indicates no pain, and 10 indicates the worst pain [23]. The Manchester Foot Pain and Disability Index (MFPDI), with good reliability (0.85) in its Spanish version, which evaluates perceived pain and disability at the foot level, was also used [24].

The Foot Function Index (FFI) [25] measures the impact of the pathology on foot functionality, considering dimensions such as pain, disability, and activity restriction. The FFI is a self-administered index consisting of 23 items divided into 3 subscales, providing total and subscale scores. Higher FFI scores indicate poorer foot health, with excellent internal consistency (0.96) in the revised Spanish version (FFI Sp-RA) and an ICC of 0.89 (95% CI: 0.84–0.92). The Manchester Scale for HAV [26] was used to assess the severity of HAV, while the Nijmegen Scale [27] was employed to estimate anatomical changes occurring in the lesser rays.

### 2.4. Procedures

Two investigators (ACC and ARC) independently interviewed the included participants to collect study data. The clinical interview was conducted in a private room at the Hospital Universitario Virgen de las Nieves. All study participants from 2018 who met the defined criteria were invited to participate again in 2023. Data collection took place between May and September. Those who agreed to participate were provided with an information sheet and an informed consent form, after which they were asked to complete the MFPDI, VAS, and FFI questionnaires. Subsequently, a foot examination was performed using the MHAV, FPI, and Nijmegen scales.

### 2.5. Statistical Analysis

Descriptive statistics (means, standard deviations, and IQRs) were calculated to characterize the data. Shapiro–Wilk tests assessed normality to determine the use of parametric (*t*-tests, ANOVA) or non-parametric (Wilcoxon, Kruskal–Wallis) tests. Post hoc analyses (Tukey or Dunn’s test) identified pairwise differences when applicable. ROC curve analysis was used to determine cutoff values for disease progression. All tests were two-tailed (*p* < 0.05) and conducted using SPSS Statistics v.26.

## 3. Results

A total of 103 patients were evaluated in this study, with a mean age of 56.88 ± 10.96 years and an average disease duration of 13.8 ± 9.42 years. In 2018, the mean height was 1.62 m, the mean weight was 71 kg, and the BMI was 29.2, improving to 26.8 in 2023. Of the participants, 75.7% were women and 24.3% were men (Table 1).

Patients in the surgery group showed relatively stable outcomes in foot function, with a mean difference between 2018 and 2023 of 2.3 ± 4.6 in total FFI and 1.9 ± 5.1 in FFI Pain, suggesting minimal changes over time. Similarly, FFI Disability remained stable, with a mean difference of 1.9 ± 6.1. In contrast, patients with ≥10 years of disease duration and non-biological treatment exhibited the largest increases in foot disability metrics; specifically, the mean difference in FFI Pain was 14.2 ± 9.6, which was significantly higher compared to the surgery group (*p* < 0.001), the FFI Disability increased by 16.1 ± 7.8 (*p* = 0.004), highlighting a greater deterioration in foot function over time. Patients treated with biological therapies generally showed moderate changes in foot function. For example, the group with ≥10 years of disease duration and biological treatment had a mean difference in FFI Pain of 9.2 ± 8.5, which was significantly lower than the corresponding non-biological group (*p* = 0.018). FFI Disability also increased less in this group (13.0 ± 7.4, *p* = 0.026) compared to the MTX ≥ 10 years group.

The surgery group exhibited better stability in pain metrics, with a mean difference of 1.8 ±6.4 in the general VAS Score, compared to a mean increase of 3.2 ± 5.9 in the MTX ≥ 10 years group (*p* = 0.014). Similarly, the Bio < 10 years group showed minimal changes in foot pain (0.6 ± 4.1, *p* = 0.212), suggesting better symptom control in this subgroup. To evaluate the effect of different treatments on disease progression, we analyzed changes in FFI between 2018 and 2023 across five treatment groups. The results revealed statistically significant differences (*p* < 0.01), indicating that treatment choice influences foot functionality over time (Table 2).

Patients with long-evolution RA (≥10 years) who did not receive biologic treatment experienced the most significant deterioration in foot function, with an average increase of 11.89 points in FFI. Patients with early-stage disease (<10 years) on non-biologic treatment were the only group to show an improvement in foot function, with a mean reduction of 5.29 points in FFI. Surgical patients exhibited only minor improvements in foot function, with an average FFI change of +2.73, but with high variability in individual outcomes. Regardless of disease duration, patients receiving biologics showed moderate changes in foot function but with substantial variability in outcomes, highlighting a heterogeneous response to biologic treatment.

Figure 1 shows the distribution of changes in general pain (VAS General) between 2018 and 2023, divided by groups.

Table 3 summarizes the main clinical predictors of disease progression from 2018 to 2023, based on a Random Forest model with permutation-based significance testing. The most influential variables included age and disease duration, followed by MFPDI, pain levels (general and foot-specific), and inflammatory activity. All listed variables were statistically significant (*p* < 0.01).

A cluster analysis based on baseline clinical variables (2018) revealed three distinct patient profiles, as shown in Table 4. Cluster 0 consisted of younger patients with low disease activity, minimal pain, and better functional outcomes (mild group). Cluster 1 presented moderate pain and disability, while Cluster 2 included older patients with high pain, structural deformities, and poorer foot function (severe group). This stratification helps illustrate the heterogeneity in disease presentation and risk of progression.

No significant differences were observed between treatment groups in the five-year evolution of general pain, FFI, or DAS28-PCR. To explore potential confounding, we conducted multivariable linear regressions, including age, disease duration, and propensity scores, none of which revealed a significant treatment effect. Additionally, stratified analyses based on baseline disease activity confirmed consistent results across subgroups. These findings suggest that similar outcomes across groups are not attributable to residual confounding but rather reflect a genuine convergence in long-term pain, function, and disease activity trajectories regardless of treatment modality.

To further evaluate the potential differences in pain progression between treatment groups, post hoc pairwise comparisons were conducted using Mann–Whitney U tests with Bonferroni correction. Despite global significance in the Kruskal–Wallis test, none of the pairwise comparisons reached statistical significance after correction. These results are presented in Appendix A and reinforce the observation that pain trajectories did not substantially differ across treatment strategies.

## 4. Discussion

The objective of the study was to analyze the postoperative evolution in patients with RA who underwent forefoot joint surgery, compared to groups of patients divided according to pharmacological treatment (MTX or biologic) with more or less than 10 years of disease progression in 2018. The study included different variables, and their evolution was observed five years later.

The results demonstrate statistically significant differences between patients who underwent forefoot surgery and those in the other comparison groups. Notably, individuals in the surgical group consistently presented with poorer outcomes in terms of pain, disability, and functional limitations. These differences persisted independently of the pharmacological treatment received. Although the primary assessment tools—VAS, MFPDI, and FFI—captured key aspects of pain and functionality, further analysis revealed that baseline systemic inflammatory activity, as measured by DAS28-PCR, was a significant predictor of progressive foot-related disability. These findings underscore the critical need for early and effective control of disease activity in patients with rheumatoid arthritis to mitigate long-term musculoskeletal sequelae.

It must be considered that patients who underwent forefoot surgery might have had a worse clinical condition at the beginning of the study. Their mean age in 2023 was 67.55 ± 6.47 years, similar to group 3 (65.09 ± 3.81) and group 4 (64.86 ± 7.29), and higher than the other groups. Regarding disease duration, the study group had a mean of 27.73 ± 8.97 years, with values comparable to group 4 (22.93 ± 3.84) and group 5 (23.81 ± 2.24). Considering the pharmacological treatment of the operated patients, 90% were receiving biologic therapy. This contrasts with the results of groups 4 and 5, where values were lower across all variables, with only a slight worsening observed after five years. In the present study, the specific surgical technique performed on each patient was not registered, as there is currently no consensus on the optimal surgical approach [28,29]. Therefore, it was assumed that each patient underwent the procedure deemed most appropriate based on their clinical condition.

At the beginning of the study, all groups showed similar mean pain levels on the VAS scale, around 5, with the highest value observed in patients who underwent forefoot surgery (5.73 ± 2.49). After five years, an increase in pain was noted, particularly in the surgically treated group and in patients with more than 10 years of disease progression treated with MTX. In contrast, both biologic treatment groups tended to maintain stable pain levels over time. Our results correlate with those of Bala et al. [30], who state that pain tends to remain stable or persistent despite current treatments that improve inflammation and continues to be a clinical problem.

When analyzing foot pain specifically using the VAS scale, all groups presented higher mean values compared to general pain levels. The surgically treated group had an initial mean score of 6.36 ± 2.93 in 2018, which increased to 6.82 ± 3.52 in 2023, approximately two points higher than the other groups. The increase in pain over five years was more significant in groups 3, 4, and 5; however, none reached the levels observed in the surgically treated group. Pharmacological treatments, including biologic disease-modifying antirheumatic drugs and conventional synthetic disease-modifying antirheumatic drugs, contribute to controlling or reducing inflammation; however, this is not always correlated with pain reduction. The increase in pain observed in patients undergoing long-term biologic and MTX treatment may be associated with the patient’s health perception, which can be influenced by factors such as osteoarticular deformities, disease duration, depression, and individual pain perception, among others [30]. In our study, surgical treatment was not effective in reducing pain, contrasting with the findings of Nomura et al., who reported that pain decreased in patients who underwent foot surgery during long-term follow-up [31].

Regarding the FFI Pain subscale, pain levels remained consistently high in the surgery group across both measurements, significantly exceeding those of the other groups. This confirms that forefoot-operated patients experience greater pain than the rest, regardless of pharmacological treatment. Pain in the foot, with current treatments, is more closely associated with deformity rather than disease activity and progression [32]. A longitudinal study with a six-year follow-up observed a high recurrence rate of HAV deformity. In cases where surgery preserved the first metatarsophalangeal joint, there was a 60% recurrence rate, while 100% of patients who underwent resection arthroplasty of all metatarsal heads experienced recurrence. Takakubo et al. [29] concluded that increased pain is directly associated with the progressive deformity of HAV. Our results support this finding, as we observed an increase in pain over time alongside a worsening HAV deviation.

On the MFPDI scale, higher values were shown in surgically treated patients, nearly twice as high as in group 2. This variable showed a slight increase across all groups after five years but without reaching the initial values observed in the surgically treated group. These values and their upward trend align with those seen in the FFI Disability subscale [33]. When analyzing functionality using the FFI and all its subscales, the study group consistently showed significantly higher values compared to the other groups. No group exhibited values as high in either 2018 or 2023. However, a general increase in pain, disability, and physical activity restriction was observed across all groups, with a particularly pronounced increase in group 3. As disease duration increases, pain tends to become more intense and frequent in patients without biologic treatment. Long-term pain is associated with increased disability, depression, and obesity [30].

In the longitudinal study by Sawachika et al. [34], an improvement in functionality was observed in patients who underwent foot surgery, with positive Health Assessment Questionnaire modified to assess disability (HAQ-DI) [35] results at 12 months. However, Kojima et al. [36] reported negative HAQ-DI outcomes in operated patients with follow-ups between 6 and 12 months. Their findings indicated that only 40% of foot surgery patients experienced functional improvement, which they attributed to good preoperative physical condition.

Moreover, Aletaha et al. [37] concluded that the HAQ-DI includes both reversible and irreversible factors, such as aging and disease progression, which could have influenced the results of both studies. This suggests that differences in patient characteristics and disease evolution may partly explain the contrasting findings.

Compared to the present results, it is important to note that the HAQ-DI, although widely used in rheumatology, is not a specific tool for evaluating foot function [35]. In contrast, the MFPDI and FFI are specifically designed to assess functional impairment in the feet of patients with RA, making them more suitable for analyzing foot-related disability [33].

The combined analysis of predictive factors and patient clustering provides a comprehensive understanding of disease progression in rheumatoid arthritis. As evidenced, variables such as age, disease duration, MFPDI, pain levels, and DAS28 were statistically significant predictors of long-term deterioration. These findings are consistent with prior research showing that advancing age, longer disease duration, and persistent inflammatory activity are associated with irreversible joint damage and functional decline [1,38]. The inclusion of both generalized and localized pain, along with foot deformity, aligns with epidemiological evidence highlighting their contribution to disability in RA patients [39].

Complementarily, the cluster analysis identified three distinct patient profiles—mild, moderate, and severe—based on these same baseline variables. Patients in the severe cluster exhibited the highest pain and disability scores, aligning with the predictors identified in the variable-level analysis. This convergence between individual predictors and group-based patterns strengthens the clinical validity of the model and supports a stratified approach to care. Such clustering strategies have been proposed as tools to enhance personalized medicine by facilitating early identification of high-risk subgroups and guiding tailored interventions [40]. To further investigate the potential influence of confounding variables, we applied several advanced statistical approaches. First, a stepwise regression analysis showed that neither age, disease duration, nor treatment group (including surgical vs. non-surgical) were significantly associated with pain evolution. The inclusion of these covariates in the model only marginally improved explanatory power (maximum R^2^ = 0.045), and the effect of surgery remained non-significant throughout.

Second, a propensity score was calculated to estimate the likelihood of undergoing surgery based on baseline characteristics. Adjusting the model using this score as a covariate also failed to identify a significant effect on pain change. Third, a stratified analysis was conducted by categorizing patients based on baseline disease activity (DAS28-PCR ≥1 vs. <1). In both strata, no significant differences were observed in pain evolution between surgical and non-surgical patients. Collectively, these findings suggest that the absence of significant differences between treatment groups is not attributable to residual confounding. Instead, they reflect a genuine convergence in pain-related outcomes across diverse treatment strategies after five years of follow-up.

It is important to acknowledge the limitations of the study. One of the main limitations is the small sample size in each group, which may limit the generalizability of the results. Furthermore, the surgery group consisted predominantly of female patients, which could introduce gender bias into the results. It should be noted that most patients in the surgery group were also receiving biological treatment, making it difficult to isolate the effects of surgery alone. This overlap reflects standard clinical treatment but may confound the interpretation of treatment-specific results. Furthermore, we decided not to include other pharmacological groups, such as tsDMARDs, avoiding greater complexity by adding other groups or combining treatments, which makes it difficult to assess their potential impact. Also, not including analgesic treatment as a variable could have been biased when analyzing pain perception in any of the study groups. Finally, we are aware that selecting patients based on their current prescribed treatment may pose a selection bias.

## 5. Conclusions

In summary, patients with RA have experienced a deterioration in terms of overall pain, foot pain, functionality, and disability between 2018 and 2023, despite surgical interventions. The present findings raise the question of whether foot surgeries, while necessary in certain cases, are sufficient to improve the clinical conditions of patients.

It is essential to adopt an interdisciplinary therapeutic approach from the early stages of the disease, incorporating specific non-surgical foot treatments and establishing more robust postoperative follow-up protocols.

## Figures and Tables

**Figure 1 healthcare-13-01004-f001:**
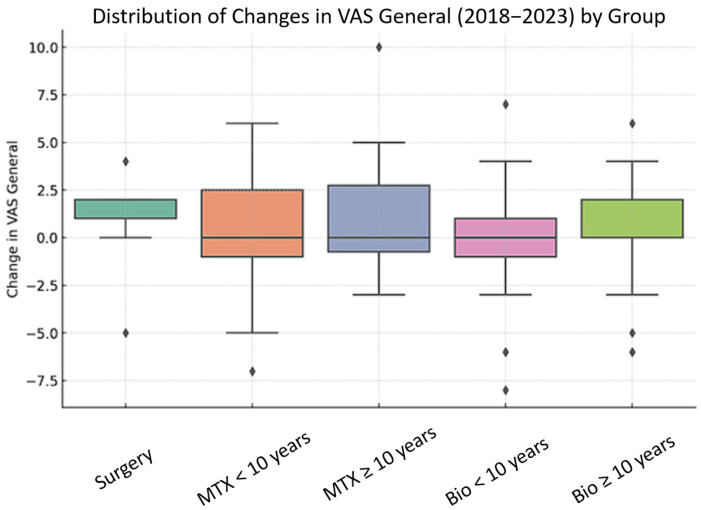
Distribution of changes in general pain (VAS General) between 2018 and 2023 by group. The analyzed groups include patients treated with surgery, methotrexate (MTX) for <10 years and ≥10 years, and biological therapy (Bio) for <10 years and ≥10 years. Each box represents the interquartile range (25–75%), with the central line indicating the median. The whiskers show values within 1.5 times the interquartile range, while the dots represent outliers. Although ANOVA and Kruskal–Wallis tests suggested global differences (*p* = 0.0089 and *p* = 0.0138, respectively), post hoc pairwise comparisons using Bonferroni-corrected Mann–Whitney U tests did not reveal any statistically significant differences between specific subgroups. A summary of pairwise comparisons is provided in Appendix A.

**Table 1 healthcare-13-01004-t001:** Descriptive characteristics of patients with rheumatoid arthritis recorded in 2018 and 2023.

Variable	Bio < 10 Years		Bio ≥ 10 Years		MTX < 10 Years		MTX ≥ 10 Years		Surgery	
*N* (patients)	20		20		20		20		20	
Age (years) 2018	58.63 ± 4.04		64.86 ± 7.29		60.76 ± 3.77		65.09 ± 3.81		67.55 ± 6.47	
Female (%)	66.6		75.6		91		71.4		91	
Disease Duration (years) 2018	9.43 ± 1.35		22.93 ± 3.84		9.73 ± 1.68		23.81 ± 2.24		27.73 ± 8.97	
General VAS Score	5.11 ±2.22	5.51 ±2.78	5.50 ±2.56	5.20 ±3.06	4.82 ±2.79	5.00 ±3.49	5.07 ±2.81	6.21 ±2.58	5.73 ±2.49	6.82 ±2.93
Feet VAS score	4.49 ±2.57	5.11 ±2.92	4.20 ±3.31	5.20 ±3.67	4.73 ±2.87	5.27 ±3.50	4.21 ±3.40	5.36 ±3.39	6.36 ±1.57	6.82 ±3.52
MFPDI	18.05 ±10.39	19.57 ±9.33	16.50 ±11.01	20.17 ±10.53	13.64 ±10.59	14.09 ±11.73	17.36 ±12.02	21.79 ±9.92	23.64 ±8.72	26.18 ±10.11
FFI	34.58 ±22.87	37.09 ±21.04	33.00 ±25.23	38.16 ±26.47	31.11 ±24.75	25.81 ±22.07	38.48 ±29.10	50.37 ±23.95	56.48 ±22.22	59.21 ±26.90
FFI Pain	37.00 ±22.99	42.43 ±23.04	33.20 ±26.91	42.60 ±28.89	33.82 ±22.79	38.00 ±27.37	36.29 ±26.06	51.64 ±25.18	59.00 ±18.11	58.09 ±25.50
FFI Disability	36.35 ±26.50	35.00 ±25.70	33.77 ±27.18	36.73 ±30.55	33.18 ±29.51	18.55 ±23.21	42.00 ±32.83	48.86 ±29.70	55.91 ±23.99	62.73 ±25.00
HAV	0.32 ±0.67	0.47 ±0.90	0.69 ±0.87	0.94 ±0.85	0.37 ±0.50	0.68 ±0.82	0.28 ±0.46	0.33 ±0.49	1.05 ±1.00	1.15 ±1.14

Values are presented as mean ± standard deviation unless otherwise indicated. Bio = biological therapy; MTX = methotrexate therapy; VAS = Visual Analogue Scale; MFPDI = Manchester Foot Pain Disability Index; FFI = Foot Function Index; HAV = Hallux Abductus Valgus (0 = No deformity, 1 = Mild deformity, 2 = Moderate deformity, 3 = Severe deformity); FPI = Foot Posture Index.

**Table 2 healthcare-13-01004-t002:** Changes in Foot Function Index between 2018 and 2023 across patient groups.

Group	Median Change	Standard Deviation	ANOVA *p*-Value
Bio < 10 years	+5.16	17.35	0.008
Bio ≥ 10 years	+2.50	21.54	
MTX < 10 years	−5.29	15.45	
MTX ≥ 10 years	+11.89	30.90	
Surgery	+2.73	20.01	

Bio = biological therapy; MTX = methotrexate therapy.

**Table 3 healthcare-13-01004-t003:** Predictive factors of disease progression (2018–2023).

Group	Importance (%)	*p*-Value
Age	22.0%	<0.001
Disease Duration	21.0%	<0.001
MFPDI	18.7%	0.007
VAS	14.0%	0.008
VAS Foot	12.1%	0.001
DAS28-PCR	12.1%	<0.001

MFPDI = Manchester Foot Pain and Disability Index; VAS = Visual Analogue Scale; DAS28-PCR = Disease Activity Score using C-reactive protein.

**Table 4 healthcare-13-01004-t004:** Clinical profile of patient clusters based on 2018 characteristics.

Cluster	Mean Age (Years)	Disease Duration (Years)	VAS Foot	MFPDI	FFI	DAS28-PCR	Cluster Interpretation
0	54.2	15.1	1.6	5.8	6.7	0.39	Mild
1	55.8	12.4	5.4	17.2	36.8	0.62	Moderate
2	60.3	14.1	6.5	28.9	63.9	0.58	Severe

FFI = Foot Function Index; MFPDI = Manchester Foot Pain and Disability Index; DAS28-PCR = Disease Activity Score using C-reactive protein.

## Data Availability

The data supporting the findings of this study are not publicly available due to privacy and ethical restrictions.

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
