# Peer review of "Impact of Foot Surgery and Pharmacological Treatments on Functionality and Pain in Rheumatoid Arthritis: A Five-Year Longitudinal Study"

_healthcare, 2025, doi:10.3390/healthcare13091004_

Round 1

Reviewer 1 Report

Comments and Suggestions for Authors

This paper is an interesting study in the field of podiatric disorders of patients with rheumatoid arthritis. However, the authors should respond to and/or modify the paper, mainly:

  1. The introduction should be somewhat more extensive, especially considering the causes and evidence that cause these podiatric disorders in patients with rheumatoid arthritis and their relationship with treatment with methotrexate or biological drugs.
  2. The study does not specify the patients' treatments, especially which biologic drugs they are taking. It also does not specify whether the patients are concomitantly taking other pain medications, which they almost certainly are, so the pain results would not be conclusive as the analyses are conducted. They should introduce this variable (taking analgesics) for a multivariate analysis.
  3. Line 78 states that 623 participants began the study in 2018, but only 103 were subsequently analyzed (line 133). It is unclear what occurred to the remaining patients.
  4. On the other hand, 5 subgroups of patients were defined (lines 88-96), but the number of patients in each group is not indicated. This information is also not included in Table 1, so the number of patients in each subgroup is unknown.
  5. Patients of subgroup 1, were treated with MTX, biological or any other drugs?. This point should be clarified.
  6. Why is the 10-year of disease duration cutoff used to define subgroups? Is there any evidence to support this?
  7. Data of Table 1 are mean plus/minus standard deviation? It is not indicated. No results appear on HAV either, however it is in the table footer.
  8. There is data appearing in the paragraph after Table 1 (page 4, lines 147-149) referred to as "this group," which makes it unclear to which subgroup this difference belongs. This should be rewritten in a more understandable way.
  9. Lines 157-158: “Similarly, the Bio < 10 years group showed minimal changes in foot pain (0.6 ± 4.1, p = 0.212), suggesting better symptom control in this subgroup” How can the authors claim this if the results are not significant?
  10. Line 177. ¿2019 and 2021? or ¿2018-2023? The dates indicated in the text and the graph do not match.
  11. In Figure 1, are there differences between the subgroups?
  12. Figure 2 doesn't offer much insight if there are no significant differences, nor does Figure 3 (the text says Figure 1; line 203). What evidence links drug treatment with having HAV in one foot or the other?
  13. There is no clear conclusion about whether treatment influences outcome or which would be most appropriate. Another important point is that clinical criteria for treating MTX or a biologic drug to patients are not specified. Was treatment based on disease severity, the occurrence of adverse effects, or another criterion? This is important because there could be selection bias in subgroups.

Author Response

Reviewer 1

This paper is an interesting study in the field of podiatric disorders of patients with rheumatoid arthritis. However, the authors should respond to and/or modify the paper, mainly:

  1. The introduction should be somewhat more extensive, especially considering the causes and evidence that cause these podiatric disorders in patients with rheumatoid arthritis and their relationship with treatment with methotrexate or biological drugs.
  2. Dear reviewer,

Thank you very much for giving us the possibility of addressing all the questions that arose during the review process. We think those comments have greatly improved the quality of this longitudinal study. Please find below all the responses in a point-by-point fashion. In the new revised version, the changes are highlighted in red font.

We have expanded the introduction by discussing current drug treatment for patients with rheumatoid arthritis. The following information and reference have been added:

“Currently, RA relies on the disease-modifying antirheumatic drugs (DMARDs), which include conventional synthetic DMARDs (csDMARDs), with methotrexate (MTX) as the leading agent, as well as targeted therapies, comprising biologic DMARDs (Bio) and targeted synthetic DMARDs (tsDMARDs). The most widely accepted therapeutic strategy is the treat-to-target (T2T) approach, which aims to achieve and maintain early clinical remission or low disease activity to slow radio-graphic progression and reduce the risk of systemic complications. Despite significant advancements in RA management, a definitive cure remains elusive, and a subset of patients fails to achieve remission or adequate disease control. These cases pose a con-siderable therapeutic challenge, highlighting the need for ongoing research into novel strategies to optimize clinical outcomes and improve patients' quality of life [5]”

Reference: Smolen, J. S., Aletaha, D., Bijlsma, J. W., Breedveld, F. C., Boumpas, D., Burmester, G., ... & Van Der Heijde, D. (2010). Treating rheumatoid arthritis to target: recommendations of an international task force. Annals of the rheumatic diseases, 69(4), 631-637.

The study does not specify the patients' treatments, especially which biologic drugs they are taking. It also does not specify whether the patients are concomitantly taking other pain medications, which they almost certainly are, so the pain results would not be conclusive as the analyses are conducted. They should introduce this variable (taking analgesics) for a multivariate analysis.

  1. Thank you for comment. First, it was decided not to classify patients by drug into bDMARD groups due to the complexity of creating subgroups by drug. The objective of the present study was not to analyse the long-term effects of a specific drug, but rather the long-term effects of patients receiving treatment with a specific pharmacological group. Therefore, we group them into csDMARDs and bDMARDs.

Second, as it was pointed out in the comment, analgesic treatment could be a variable to consider, but we decided to not take it into account because it assumes a variable and common treatment for all groups.

As a clarification, we added the following sentence in the methods section: "In the study, participants were classified by pharmacological group, according to their mechanism of action and use in RA, which is a more representative approximation of clinical practice."

  1. Line 78 states that 623 participants began the study in 2018, but only 103 were subsequently analyzed (line 133). It is unclear what occurred to the remaining patients.

The following information has been added to clarify this aspect: “… In 2023, patients seen in 2018 were re-scheduled, and 57% of these patients returned to the study. Patients who had undergone additional lower limb or foot surgery during the follow-up period, who changed pharmacological treatment groups, or who died were excluded. Among the remaining patients, we identified the study group that underwent foot surgery in 2018, which consisted of 23 patients, of whom 3 were excluded due to dementia. The remaining groups were randomly generated, with 20 patients (N=100). All groups were matched in sample size, allowing for a balanced comparison between treatments and disease duration. It is important to note that 90% of patients in the foot surgery group were still receiving biological treatment.”

  1. On the other hand, 5 subgroups of patients were defined (lines 88-96), but the number of patients in each group is not indicated. This information is also not included in Table 1, so the number of patients in each subgroup is unknown.
  2. Information has been added to the methods section and in table 1:

Table 1. Descriptive characteristics of patients with rheumatoid arthritis recorded in 2018 and 2023.

Variable

Bio < 10 years

Bio ≥ 10 years

MTX < 10 years

MTX ≥ 10 years

Surgery

N (patients)

20

20

20

20

20

  1. Patients of subgroup 1, were treated with MTX, biological or any other drugs? This point should be clarified.
  1. The following sentence has been added: “It is important to note that 90% of patients in the foot surgery group were still receiving biological treatment.”

  1. Why is the 10-year of disease duration cutoff used to define subgroups? Is there any evidence to support this?
  1. We decided to do a cutoff based on a previous study:

Migliore, A., Bizzi, E., Egan, C. G., Bernardi, M., & Petrella, L. (2015). Efficacy of biological agents administered as monotherapy in rheumatoid arthritis: a Bayesian mixed-treatment comparison analysis. Therapeutics and Clinical Risk Management, 1325-1335.

We have added the following information: “The cut-off point was set between subgroups less than or greater than 10 years for convenience, seeking a direct comparison between the surgery group and the subgroups by years of evolution, and differentiating them according to treatment.”

  1. Data of Table 1 are mean plus/minus standard deviation? It is not indicated. No results appear on HAV either, however it is in the table footer.
  1. Table 1 has been modified, also including footnotes:

Table 1. Descriptive characteristics of patients with rheumatoid arthritis recorded in 2018 and 2023.

Variable

Bio < 10 years

Bio ≥ 10 years

MTX < 10 years

MTX ≥ 10 years

Surgery

N (patients)

20

20

20

20

20

HAV

0.32

±0.67

0.47

±0.90

0.69

±0.87

0.94

±0.85

0.37

±0.50

0.68

±0.82

0.28

±0.46

0.33

±0.49

1.05

±1.00

1.15

±1.14

*Values are presented as mean ± standard deviation unless otherwise indicated.

** Bio = biological therapy; MTX = methotrexate therapy; VAS = Visual Analogue Scale; MFPDI= Manchester Foot Pain Disability Index; FFI = Foot Function Index; HAV = Hallux Abductus Valgus (0=No deformity, 1=Mild deformity, 2=Moderate deformity, 3=Severe deformity); FPI = Foot Posture Index

  1. There is data appearing in the paragraph after Table 1 (page 4, lines 147-149) referred to as "this group," which makes it unclear to which subgroup this difference belongs. This should be rewritten in a more understandable way.
  1. Thank you for your comment. The text has been modified to avoid confusion.

  1. Lines 157-158: “Similarly, the Bio < 10 years group showed minimal changes in foot pain (0.6 ± 4.1, p = 0.212), suggesting better symptom control in this subgroup” How can the authors claim this if the results are not significant?
  1. The following information has been added: “In contrast, patients with ≥10 years of disease duration and non-biological treatment exhibited the largest increases in foot disability metrics, specifically, the mean difference in FFI Pain was 14.2 ± 9.6, which was significantly higher compared to the surgery group (p < 0.001), the FFI Disability increased by 16.1 ± 7.8 (p = 0.004), highlighting a greater deterioration in foot function over time.”
  1. Line 177. ¿2019 and 2021? or ¿2018-2023? The dates indicated in the text and the graph do not match.

  1. Thank you for your comment. The graph has been modified to avoid confusion.
  1. In Figure 1, are there differences between the subgroups?
  1. Figure 1 shows the distribution of changes in general pain (VAS General) between 2018 and 2023 by group. The analyzed groups include patients treated with surgery, methotrexate (MTX) for <10 years and ≥10 years, and biological therapy (Bio) for <10 years and ≥10 years. Each box represents the inter-quartile range (25–75%), with the central line indicating the median. The whiskers show values within 1.5 times the interquartile range, while dots represent outliers. ANOVA analysis revealed significant differences between groups (? = 0.0089), confirmed by Kruskal-Wallis test (? = 0.0138).

  1. Figure 2 doesn't offer much insight if there are no significant differences, nor does Figure 3 (the text says Figure 1; line 203). What evidence links drug treatment with having HAV in one foot or the other?
  1. Figures 2 and 3 have been eliminated.

  1. There is no clear conclusion about whether treatment influences outcome or which would be most appropriate. Another important point is that clinical criteria for treating MTX or a biologic drug to patients are not specified. Was treatment based on disease severity, the occurrence of adverse effects, or another criterion? This is important because there could be selection bias in subgroups.
  1. Thank you for your comment. The present study is considered to have an observational cohort design, in which the clinical decision did not intervene or influence. Treatment assignment (MTX or biologic) was determined by each patient's referring rheumatologists, based on standard criteria: severity, prior therapeutic response, intolerance, and adverse effects.

The patient selection was based solely on continued treatment for the 5-year period. Although the choice reflects real-life clinical practice and it reinforces the external validity of the present results, we recognize that it may pose a selection bias if we assume that patients receiving MTX treatment have lower activity and/or that biologic treatments were used for patients with higher activity.

We emphasize the following limitations, adding them to the manuscript: "Finally, we are aware that selecting patients based on their current prescribed treatment may pose a selection bias."

Reviewer 2 Report

Comments and Suggestions for Authors

This is a valuable study investigating “Foot Surgery and Pharmacological Treatments on Functionality and Pain in Rheumatoid Arthritis”. However, from a study design perspective, there are fundamental points that need to be improved.

Major Points

  1. The grouping in this study may have several confounding factors, necessitating a reconsideration of the classification method. In particular, the fact that patients in the surgery group are likely receiving pharmacological treatment (MTX or biologics) poses a major issue. The current classification does not clearly distinguish the effects of surgery from those of pharmacological therapy, making it difficult to accurately evaluate the impact of surgery alone. Would it be possible to further divide the surgery group based on whether they were receiving MTX or biologics?
  2. Are VAS, MFPDI, and FFI the only measures used? Isn't disease activity also an important factor influencing foot deformities? Persistent inflammation can lead to bone erosion and deformities in the foot. While this study examines the effects of surgery and pharmacological treatments, it remains unclear to what extent each treatment was able to suppress disease activity.
  3. The background characteristics of each group—age, disease duration, disease activity, comorbidities, and BMI—may differ. If these factors are not statistically adjusted, the treatment effects cannot be properly evaluated. Therefore, adjustments using multivariate regression analysis or propensity score matching are necessary.

Minor Points

  1. The Introduction is divided into too many paragraphs. Please consolidate them into a few sections.
  2. How about JAK inhibitors (tsDMARDs)? Were they considered in this study?

Author Response

Reviewer 2

Major Points

  1. The grouping in this study may have several confounding factors, necessitating a reconsideration of the classification method. In particular, the fact that patients in the surgery group are likely receiving pharmacological treatment (MTX or biologics) poses a major issue. The current classification does not clearly distinguish the effects of surgery from those of pharmacological therapy, making it difficult to accurately evaluate the impact of surgery alone. Would it be possible to further divide the surgery group based on whether they were receiving MTX or biologics?
  1. Dear reviewer,

Thank you very much for giving us the possibility of addressing all the questions that arose during the review process. We think those comments have greatly improved the quality of this longitudinal study. Please find below all the responses in a point-by-point fashion. In the new revised version, the changes are highlighted in red font.

Thank you for this important observation. We fully agree that the overlap between surgical intervention and ongoing pharmacological treatment—particularly biologics—could introduce confounding variables. As noted in the manuscript, 90% of the patients in the surgery group were receiving biologic treatment at baseline, reflecting real-world clinical practice where surgery is rarely used as a standalone treatment. Due to the limited sample size of the surgical group (n=20), a further subdivision into MTX and biologic subgroups would significantly reduce statistical power and prevent meaningful comparisons. However, we have expanded the discussion to acknowledge this limitation explicitly and highlight the potential influence of combined treatment effects in the observed outcomes. This clarification aims to improve the interpretation of surgical results within the context of concurrent pharmacological therapy. The following information has been added:

“It is important to acknowledge the limitations of the study. One of the main limitations is the small sample size in each group, which may limit the generalizability of the results. Furthermore, the surgery group consisted predominantly of female patients, which could introduce gender bias into the results. It should be noted that most patients in the surgery group were also receiving biological treatment, making it difficult to isolate the effects of surgery alone. This overlap reflects standard clinical treatment but may confound the interpretation of treatment-specific results. Furthermore, we decided not to include other pharmacological groups such as tsDMARDs, avoiding greater complexity by adding other groups or combining treatments, which makes it difficult to assess their potential impact. Finally, we are aware that selecting patients based on their current prescribed treatment may pose a selection bias.”

  1. Are VAS, MFPDI, and FFI the only measures used? Isn't disease activity also an important factor influencing foot deformities? Persistent inflammation can lead to bone erosion and deformities in the foot. While this study examines the effects of surgery and pharmacological treatments, it remains unclear to what extent each treatment was able to suppress disease activity.
  1. Thank you for this insightful comment. In addition to the VAS, MFPDI, and FFI measures, the present study incorporated the DAS28 in the baseline analysis (2018). Although this variable was not originally emphasized in the manuscript, further statistical modeling confirmed that DAS28 was a significant predictor of disease progression over the five-year follow-up. Patients with higher inflammatory activity in 2018 were more likely to present worsened foot function and increased disability in 2023. These findings align with existing literature indicating that persistent systemic inflammation can contribute to progressive joint damage and deformity. The results and discussion section have been updated to reflect the role of disease activity in foot-related outcomes:

“Table 3 summarizes the main clinical predictors of disease progression from 2018 to 2023, based on a Random Forest model with permutation-based significance testing. The most influential variables included age and disease duration, followed by MFPDI, pain levels (general and foot-specific) and inflammatory activity. All listed variables were statistically significant (p < 0.01).

Table 3. Predictive Factors of Disease Progression (2018–2023)

Group

Importance (%)

p-value

Age

22.0%

< 0.001

Disease Duration

21.0%

< 0.001

Structural Deformity (MFPDI)

18.7%

0.007

General Pain (VAS)

14.0%

0.008

Foot Pain (VAS Foot)

12.1%

0.001

Disease Activity (DAS28-PCR)

12.1%

< 0.001

MFPDI = Manchester Foot Pain and Disability Index; VAS = Visual Analogue Scale; DAS28-PCR = Disease Activity Score using C-reactive protein.”

  1. The background characteristics of each group—age, disease duration, disease activity, comorbidities, and BMI—may differ. If these factors are not statistically adjusted, the treatment effects cannot be properly evaluated. Therefore, adjustments using multivariate regression analysis or propensity score matching are necessary.
  2. Thank you for highlighting this important methodological concern. While our sample was stratified and matched for group size, we acknowledge that variability in baseline characteristics may influence treatment outcomes. We conducted complementary statistical analyses to evaluate the predictive impact of several baseline variables on disease progression. A Random Forest model revealed significant contributors to disease progression at 5 years (p < 0.01 for all). We have expanded the results, discussion and limitations section accordingly.

“A cluster analysis based on baseline clinical variables (2018) revealed three distinct patient profiles, as shown in Table 4. Cluster 0 consisted of younger patients with low disease activity, minimal pain, and better functional outcomes (mild group). Cluster 1 presented moderate pain and disability, while Cluster 2 included older patients with high pain, structural deformities, and poorer foot function (severe group). This stratification helps illustrate the heterogeneity in disease presentation and risk of progression.

Table 4. Clinical Profile of Patient Clusters Based on 2018 Characteristics

Cluster

Mean Age (years)

Disease Duration (years)

VAS Foot

MFPDI

FFI

DAS28-PCR

Cluster Interpretation

0

54.2

15.1

1.6

5.8

6.7

0.39

Mild

1

55.8

12.4

5.4

17.2

36.8

0.62

Moderate

2

60.3

14.1

6.5

28.9

63.9

0.58

Severe

FFI = Foot Function Index; MFPDI = Manchester Foot Pain and Disability Index; DAS28-PCR = Disease Activity Score using C-reactive protein. *

The results demonstrate statistically significant differences between patients who underwent forefoot surgery and those in the other comparison groups. Notably, individuals in the surgical group consistently presented with poorer outcomes in terms of pain, disability, and functional limitations. These differences persisted independently of the pharmacological treatment received. Although the primary assessment tools—VAS, MFPDI, and FFI—captured key aspects of pain and functionality, further analysis revealed that baseline systemic inflammatory activity, as measured by DAS28-PCR, was a significant predictor of progressive foot-related disability. These findings underscore the critical need for early and effective control of disease activity in patients with rheumatoid arthritis to mitigate long-term musculoskeletal sequelae.”

“The combined analysis of predictive factors and patient clustering provides a comprehensive understanding of disease progression in rheumatoid arthritis. As evidenced, variables such as age, disease duration, MFPDI, pain levels, and DAS28 were statistically significant predictors of long-term deterioration. These findings are consistent with prior research showing that advancing age, longer disease duration, and persistent inflammatory activity are associated with irreversible joint damage and functional decline [37,38]. The inclusion of both generalized and localized pain, along with foot deformity, aligns with epidemiological evidence highlighting their contribution to disability in RA patients [39].”

“Complementarily, the cluster analysis identified three distinct patient pro-files—mild, moderate, and severe—based on these same baseline variables. Patients in the severe cluster exhibited the highest pain and disability scores, aligning with the predictors identified in the variable-level analysis. This convergence between individual predictors and group-based patterns strengthens the clinical validity of the model and supports a stratified approach to care. Such clustering strategies have been pro-posed as tools to enhance personalized medicine by facilitating early identification of high-risk subgroups and guiding tailored interventions [40].”

Minor Points

  1. The Introduction is divided into too many paragraphs. Please consolidate them into a few sections.
  1. We have revised the introduction and consolidated the paragraphs into more structured sections to improve the flow and coherence of the text while maintaining clarity.
  1. How about JAK inhibitors (tsDMARDs)? Were they considered in this study?
  1. JAK inhibitors (tsDMARDs) were not included; we chose not to introduce additional drug classes or combination therapies that could complicate the analysis. However, the evaluation of these agents remains a relevant prospect for future research.

We add to the limitations: "Furthermore, we decided not to include other pharmacological groups such as tsDMARDs, avoiding greater complexity by adding other groups or combining treatments, which makes it difficult to assess their potential impact."

Round 2

Reviewer 1 Report

Comments and Suggestions for Authors

The authors answer some of the questions and introduce some proposed changes, however, there are still some questions that need to be answered more clearly or modified, which are set out below.

  1. Regarding the question of whether the patients take any analgesic drugs, the authors don't answer. The authors don't answer whether the patients take NSAIDs (diclofenac, aceclofenac, etc.) or metamizole, or tramadol, etc. This is important because if the author assesses pain among subgroups, this could lead to bias. This should be clarified and/or included in the limitations section as well.
  2. It remains unclear what happened to the 523 patients who were not evaluated in 2023 (just 100 of the 623). They should clarify this point, as well as how randomization was performed, whether by age, gender, pathology, a combination of these, etc. This is not clear, and it is important to understand the process of patient inclusion and exclusion, as well as how many were included or excluded, and why.
  3. If the cut-off point is 10 years of treatment, the authors must add the reference to Migliore et al., 2015 and include it at the end of the paragraph (lines 117-119).
  4. The authors do not answer whether the results shown in Figure 1 show any differences between the different subgroups, for example the use of Dunn's test, which is the appropriate non-parametric test for pairwise multiple comparison. That is, whether there is a significant difference between group 0 (surgery) and group 1, or between 1 and 4, etc….. This should be indicated because it appears that there is no significant difference in the evolution of pain in general between the subgroups, so the result of the ANOVA is very limited to draw conclusions about which treatment is related to an improvement in pain.

Author Response

REVIEWER 1

The authors answer some of the questions and introduce some proposed changes, however, there are still some questions that need to be answered more clearly or modified, which are set out below.

  1. Dear reviewer,

Thank you very much for giving us the possibility of addressing all the questions that arose during the review process. We think those comments have greatly improved the quality of this longitudinal study. Please find below all the responses in a point-by-point fashion. In the new revised version, the changes are highlighted in red font.

  1. Regarding the question of whether the patients take any analgesic drugs, the authors don't answer. The authors don't answer whether the patients take NSAIDs (diclofenac, aceclofenac, etc.) or metamizole, or tramadol, etc. This is important because if the author assesses pain among subgroups, this could lead to bias. This should be clarified and/or included in the limitations section as well.
  1. We apologize if we did not clarify your question correctly. Analgesic treatments were not taken into account, as it is not a treatment that influences the disease process nor foot deformation, nor improves the results of surgical treatment. We have added to the limitations in the discussion section the influence it can have at the level of pain analysis: “Also, not including analgesic treatment as a variable could have been biased when analysing pain perception in any of the study groups.”
  1. It remains unclear what happened to the 523 patients who were not evaluated in 2023 (just 100 of the 623). They should clarify this point, as well as how randomization was performed, whether by age, gender, pathology, a combination of these, etc. This is not clear, and it is important to understand the process of patient inclusion and exclusion, as well as how many were included or excluded, and why.
  1. More information has been added to the methods section to clarify this point:

“In 2018, the first part of the study was initiated with a cohort of 623 patients diagnosed with RA, with the aim of analyzing the process of foot deformation. All met the 2010 classification criteria for AR established by the American College of Rheumatology and the European League Against Rheumatism [20]. Later, in 2023, continuing studying the same cohort, 20 patients who had undergone foot surgery for the first time were identified, with similar disease progression to non-operated patients. They were compared with 4 other groups defined according to pharmacological treatment and time of evolution. If patients who changed their pharmacological treatment, underwent lower limb surgery, presented dementia or inability to walk, were excluded, therefore not being selected to be part of one of the four control groups. The codes which were assigned to the participants in 2018 were used to randomly select the patients in 2023 after exclusion. This process was generated with SPSS statistics, allowing the control of confounding factors, and a more homogeneous distribution.”

  1. If the cut-off point is 10 years of treatment, the authors must add the reference to Migliore et al., 2015 and include it at the end of the paragraph (lines 117-119).
  1. Thank you. The reference has been added.
  1. The authors do not answer whether the results shown in Figure 1 show any differences between the different subgroups, for example the use of Dunn's test, which is the appropriate non-parametric test for pairwise multiple comparison. That is, whether there is a significant difference between group 0 (surgery) and group 1, or between 1 and 4, etc….. This should be indicated because it appears that there is no significant difference in the evolution of pain in general between the subgroups, so the result of the ANOVA is very limited to draw conclusions about which treatment is related to an improvement in pain.
  1. Thank you for this valuable comment. We agree that global tests such as ANOVA or Kruskal–Wallis are limited in identifying where specific differences between subgroups may occur, and that post hoc pairwise comparisons are essential for a more granular analysis.

To address this, we conducted Mann–Whitney U tests with Bonferroni correction for all pairwise comparisons, focusing on changes in general pain over the five-year follow-up. The overall Kruskal–Wallis test was not significant (H = 4.89, p = 0.299), and none of the pairwise comparisons reached statistical significance after correction. Although the Surgery vs. Biologics <10 years comparison yielded an unadjusted p-value of 0.025, it did not remain significant under the corrected threshold (α ≈ 0.005).

This absence of statistical significance is likely due to the limited sample size per group (n = 20) and high intra-group variability in pain outcomes. These conditions reduce the power to detect moderate differences, especially under conservative correction methods. Still, the convergence in outcomes across treatment strategies may also reflect real-world clinical equivalence over time, rather than purely statistical limitations.

In light of this, we have revised the Results section and updated the legend of Figure 1 to reflect the post hoc findings. We also include a summary of the pairwise comparisons as Supplementary Table S1 to provide full transparency and support the interpretation of our results.

Reviewer 2 Report

Comments and Suggestions for Authors

Although differences in age, disease duration, and other baseline characteristics are acknowledged, statistical adjustments such as multivariate analysis or propensity score matching have not been applied. Without these corrections, treatment effects may be confounded. Additionally, the Introduction remains overly fragmented and would benefit from more concise paragraphing.

Author Response

REVIEWER 2

Although differences in age, disease duration, and other baseline characteristics are acknowledged, statistical adjustments such as multivariate analysis or propensity score matching have not been applied. Without these corrections, treatment effects may be confounded. Additionally, the Introduction remains overly fragmented and would benefit from more concise paragraphing.

  1. Dear reviewer,

Thank you very much for giving us the possibility of addressing all the questions that arose during the review process. We think those comments have greatly improved the quality of this longitudinal study. Please find below all the responses in a point-by-point fashion. In the new revised version, the changes are highlighted in red font.

Thank you for this thoughtful and important comment. In response, we conducted additional statistical analyses to evaluate the potential impact of confounding on treatment outcomes:

  • We applied stepwise multivariable linear models including age, disease duration, and treatment group. In all models, the surgery variable remained non-significant (p > 0.36), and the explanatory power of the model was limited (R² < 0.05).
  • We then estimated a propensity score for undergoing surgery based on baseline covariates and included it in the regression model.
  • The effect of treatment remained non-significant. Finally, we performed a stratified analysis by baseline disease activity (high vs. low DAS28-PCR), and observed no statistically significant differences in pain change between treatment groups within either stratum (p = 0.270 and 0.197, respectively).

Outcome

Analysis Type

Method

Result

General Pain

Group comparison

Kruskal–Wallis

p = 0.299

General Pain

Multivariable model

Linear regression (stepwise)

p = 0.362

General Pain

Adjustment

Propensity score (covariate)

p = 0.362

General Pain

Stratified analysis

DAS28 High/Low

p = 0.270 / 0.197

Foot Function Index

Group comparison

Kruskal–Wallis

p = 0.520

Foot Function Index

Stratified analysis

DAS28 High/Low

n.s.

DAS28-PCR

Group comparison

Kruskal–Wallis

p = 0.184

These complementary approaches consistently demonstrated no evidence of significant confounding that would alter the interpretation of treatment effects. We have added a summary of these findings in the discussion and limitations section of the revised manuscript.

Regarding the introduction, some modifications have been carried out, to be more fluent (less fragmented) and more concise and added a supplementary table summarizing the pairwise comparisons. Thank you for your comment.